# The Integration of Triboelectric Nanogenerators and Supercapacitors: The Key Role of Cellular Materials

**DOI:** 10.3390/ma16103751

**Published:** 2023-05-15

**Authors:** Jiajing Meng, Zequan Zhao, Xia Cao, Ning Wang

**Affiliations:** 1Center for Green Innovation, School of Mathematics and Physics, University of Science and Technology Beijing, Beijing 100083, China; s20200767@xs.ustb.edu.cn (J.M.); m202110789@xs.ustb.edu.cn (Z.Z.); 2Beijing Institute of Nanoenergy and Nanosystems, Chinese Academy of Sciences, Beijing 100083, China; 3School of Chemistry and Biological Engineering, University of Science and Technology Beijing, Beijing 100083, China

**Keywords:** supercapacitors, triboelectric nanogenerator, cellular materials, energy harvesting and storage

## Abstract

The growing demand for sustainable and efficient energy harvesting and storage technologies has spurred interest in the integration of triboelectric nanogenerators (TENGs) with supercapacitors (SCs). This combination offers a promising solution for powering Internet of Things (IoT) devices and other low−power applications by utilizing ambient mechanical energy. Cellular materials, featuring unique structural characteristics such as high surface−to−volume ratios, mechanical compliance, and customizable properties, have emerged as essential components in this integration, enabling the improved performance and efficiency of TENG−SC systems. In this paper, we discuss the key role of cellular materials in enhancing TENG−SC systems’ performance through their influence on contact area, mechanical compliance, weight, and energy absorption. We highlight the benefits of cellular materials, including increased charge generation, optimized energy conversion efficiency, and adaptability to various mechanical sources. Furthermore, we explore the potential for lightweight, low−cost, and customizable cellular materials to expand the applicability of TENG−SC systems in wearable and portable devices. Finally, we examine the dual effect of cellular materials’ damping and energy absorption properties, emphasizing their potential to protect TENGs from damage and increase overall system efficiency. This comprehensive overview of the role of cellular materials in the integration of TENG−SC aims to provide insights into the development of next−generation sustainable energy harvesting and storage solutions for IoT and other low−power applications.

## 1. Introduction

Triboelectric nanogenerators (TENGs) have been at the forefront of renewable energy harvesting since their invention in 2012, providing a viable means of converting various mechanical energies present in the ambient environment into electrical energy [1,2,3,4,5,6,7,8,9,10,11]. Their benefits include their environmental friendliness, low cost, high efficiency, and wide range of material choices, making them suitable for numerous self−powered electronic devices. However, the unstable and irregular nature of mechanical energy sources, coupled with the high voltage and high impedance output of TENGs, poses challenges for their direct utilization as power sources.

Meanwhile, supercapacitors store and release energy through electrochemical interactions between ions and electrodes. They offer fast charging and discharging times, high power density, and long cycle life, making them suitable for integration with energy harvesting devices such as TENGs. Nevertheless, the energy storage capacity of supercapacitors still needs improvement to meet the demands of various applications.

Cellular materials, including foams, porous structures, and scaffolds, have emerged as promising solutions to address the limitations of both TENGs and supercapacitors. These materials offer several advantages, such as increased surface area, improved mechanical properties, and enhanced ion transport, which can significantly improve the performance of TENGs and supercapacitors.

For TENGs, cellular materials can provide a larger effective contact area between two triboelectric materials, enhancing the amount of charge generated during the triboelectric process and resulting in higher output power. Moreover, the flexible nature of cellular materials enables the development of wearable and soft electronics applications, while their lightweight and compressible properties are advantageous for energy harvesting applications with weight and size constraints.

In the case of supercapacitors, cellular materials can increase the surface area for efficient charge storage, directly impacting the capacitance of the device. The porous nature of these materials also facilitates faster ion transport, leading to improved charging and discharging times. Furthermore, cellular materials can provide mechanical stability to the electrodes, making them more resistant to deformation during repeated charge–discharge cycles and extending the device’s lifespan.

The integration of TENGs and supercapacitors, aided by cellular materials, has the potential to create a self−charging, uninterrupted power supply (TENG−UPS) system. Compensating for energy consumption and extending the working duration of electronic devices’ cellular materials have proven to be essential in the integration of TENGs and supercapacitors, providing a viable solution for developing efficient, lightweight, and flexible energy harvesting and storage devices (Figure 1). As research in this field progresses, there is a need to address the challenges and opportunities associated with the integration of TENGs and supercapacitors, paving the way for a more sustainable future for IoT devices and other electronic systems.

In this review, we will explore the recent advancements in the integration of TENGs and supercapacitors with cellular materials. We will discuss the benefits of using cellular materials in TENGs and supercapacitors and how they can improve their performance. Furthermore, we will delve into the challenges associated with the integration of TENGs and supercapacitors and how cellular materials can address them. Finally, we will highlight the potential applications of TENG−UPS systems and future research directions in this field. 

## 2. Operating Principle

### 2.1. Working Principle of TENGs

TENGs are a cutting−edge technology that converts mechanical energy into electrical energy by exploiting the combined effects of the triboelectric effect and electrostatic induction. When two dissimilar materials with different electron affinities come into contact, they exchange electrons, creating opposite charges on their surfaces [12,13,14,15,16,17,18,19]. The repeated contact–separation or relative sliding motion of the materials generates an alternating current (AC) output. TENGs can be classified into four distinct working modes: the vertical contact–separation mode, the lateral sliding mode, the single−electrode mode, and the freestanding triboelectric layer mode (Figure 2).

Vertical Contact–Separation (CS) Mode: In this mode, two materials with different electron affinities come into contact, exchanging electrons and generating opposite charges on their surfaces. As the materials separate vertically, electrostatic induction occurs, causing a potential difference between their electrodes and producing an electric current. The repetition of the contact–separation cycles results in an AC output.

Lateral Sliding (LS) Mode: Similar to the vertical contact–separation mode, the lateral sliding mode involves the horizontal displacement of two materials. The alternating current is generated by repeatedly sliding the materials against each other.

Single−Electrode (SE) Mode: The single−electrode mode employs the Earth as a reference electrode. The potential difference between the metal electrode and the Earth generates a current due to electrostatic induction when the triboelectric material comes into contact with or separates from the Earth’s surface.

Freestanding Triboelectric Layer (FT) Mode: In this mode, a charged object is placed between two electrodes attached to a dielectric layer. The movement of the charged object between the two electrodes alters the potential difference, generating an electric current.

TENGs can be customized to meet specific application requirements by selecting various friction layer materials and electrode materials. With their high−power output capabilities, TENG−based devices can support complex functional modules, making them suitable for a wide range of applications, including medical devices.

### 2.2. Working Principle of Supercapacitors

Supercapacitors, also known as ultracapacitors or electrochemical capacitors, are energy storage devices that bridge the gap between conventional capacitors and batteries. They possess high power density, fast charging and discharging capabilities, and a long cycle life, making them suitable for various applications, including integration with energy harvesting devices such as TENGs [21,22,23,24].

The working principle of supercapacitors is based on the electrostatic interactions between charged particles (ions) and electrodes. Unlike batteries, which rely on chemical reactions for energy storage, supercapacitors store energy through physical processes, allowing for faster and more efficient charge and discharge cycles.

There are two primary types of supercapacitors: electric double−layer capacitors (EDLCs) and pseudocapacitors. The energy storage mechanisms in these supercapacitors are distinct, although their overall operation follows a similar principle.

Electric Double−Layer Capacitors (EDLCs): EDLCs store energy through the formation of an electric double layer at the interface between the electrode and the electrolyte. When a voltage is applied, ions from the electrolyte accumulate on the surface of the electrode, creating an electric double layer that stores energy in the form of electrostatic charges. The capacitance of an EDLC is primarily determined by the surface area of the electrodes and the thickness of the electric double layer.

Pseudocapacitors: Pseudocapacitors store energy through a combination of electric double−layer capacitance and fast, reversible faradaic (redox) reactions that occur on the surface of the electrodes. The electrode materials in pseudocapacitors are typically transition metal oxides or conducting polymers, which exhibit high specific capacitances due to their fast and reversible redox reactions. Pseudocapacitors generally have higher energy densities than EDLCs but may have lower power densities and shorter cycle lives.

In both types of supercapacitors, the energy storage capacity is directly related to the surface area of the electrodes, the distance between the electrode and the electrolyte ions, and the dielectric constant of the electrolyte. By optimizing these factors, researchers can enhance the performance of supercapacitors, making them an attractive option for energy storage in various applications, including their integration with TENGs for the development of TENG−UPS systems.

### 2.3. Working Principle of TENG−SCs

The working principle of TENG−SCs involves the conversion of the irregular and intermittent AC output from the TENG into a stable DC output capable of powering electronic devices. This is achieved by passing the AC output through a rectifier bridge, which converts it into a direct current. The DC output is then stored in an electrochemical energy storage unit, such as a supercapacitor or a battery.

As external forces act upon the TENG−SC system, the energy storage unit accumulates more energy. This stored energy can then be used to charge and discharge the TENG−SC when connected to an external load. This process results in a stable DC power source that can provide a continuous and adjustable output for powering electronic devices.

By combining TENGs with supercapacitors, the TENG−SC system offers a reliable and sustainable energy solution that can be utilized in various applications. The system is capable of efficiently harvesting and storing energy from mechanical sources, while also providing a stable DC output for electronic devices. As such, TENG−SC technology represents a promising direction for the advancement of energy harvesting and storage technologies.

## 3. Cellular Materials in TENG

### 3.1. Cellulose in TENG

#### 3.1.1. Natural Cellulose 

Natural cellulose in TENG applications is derived from a renewable and eco−friendly biopolymer that possesses distinct properties such as hydrophilicity, biodegradability, and mechanical stability. These unique characteristics make natural cellulose a promising candidate for TENG applications. Researchers have worked with various cellulose−based materials, including cellulose templates and cellulose nanofibers, that demonstrate the potential for enhanced performance in energy harvesting, humidity sensing, and respiration monitoring. The natural structure of cellulose allows for increased contact area and charge density, contributing to advancements in TENG technology.

Zhang et al. designed a hydrophilic triboelectric material using a natural cellulose template and Ti_3_C_2_T_x_ nanosheets, enhancing its sensitivity, stability, and wireless transmission capabilities for self−powered sensing in high−humidity environments (Figure 3a) [25]. The cellulose template, with its porous honeycomb structure and abundant hydroxyl groups, facilitates the attachment of Ti_3_C_2_T_x_ and improves the material’s mechanical stability and hydrophilicity. This interconnected network offers numerous adsorption sites for water molecules, resulting in excellent performance over a wide humidity range (40%–90% RH), high sensitivity (0.8/1%), and good stability (150 s). The cellulose−based triboelectric material enables non−contact humidity sensing and respiration monitoring, demonstrating great potential for self−powered wireless sensors in various applications. This innovative approach to designing high−performance triboelectric materials using cellulose templates opens up new possibilities for the development and utilization of cellulose−based materials in self−powered sensing technologies. 

In the study conducted by Pongsakorn et al., a natural rubber (NR)−based TENG is designed, leveraging the porous structure of cellulose nanofibers (CNF) and activated carbon (AC) nanoparticles to enhance its power output (Figure 3b) [26]. The CNF, extracted from agricultural waste, increases the positive tribopolarity of NR, while the AC, with its large specific surface area, improves charge generation and storage. The researchers experimented with various filler concentrations and compared the performance of the NR−based TENGs with a pure NR TENG and a commercial PTFE−based TENG. The NR−CNF−AC TENG containing 10wt% CNF and 5wt% AC fillers achieved the highest performance, with a voltage output of 137 V, a current of 12.1 µA, and a power density of 2.74 W m^−2^. The results demonstrate that the combination of CNF and AC has a synergistic effect on enhancing the TENG’s output, highlighting the potential for using natural−based materials for energy harvesting. The NR−based TENG can also charge capacitors and power LEDs using a rectifier circuit, indicating its potential as an eco−friendly power source for micro/nano−electronic devices. 

#### 3.1.2. Modified Cellulose 

Modified cellulose in TENG applications involves altering the natural cellulose structure to improve its functionality and performance. Examples of modified cellulose materials include cyanoethyl cellulose and other chemically treated cellulose derivatives. By modifying cellulose, researchers can achieve improved hydrophilicity, increased contact area, and enhanced charge density [30,31,32,33,34,35,36,37,38,39,40,41]. As a result, these modified cellulose materials unlock new possibilities for TENG applications, such as temperature sensing, flame retardancy, and more efficient energy conversion. The development of modified cellulose materials continues to push the boundaries of TENG technology, offering innovative and sustainable solutions for energy harvesting and sensing applications.

Wang et al. have developed a modified cellulose material, cyanoethyl cellulose (CE−cellulose), that enhances the electrical output of TENGs in high humidity environments [27]. The cellular structure of CE−cellulose allows it to be lightweight and flexible, making it ideal for energy harvesting and storage devices. CE−cellulose exhibits improved contact electrification performance when in contact with different electrode materials, such as polytetrafluoroethylene (PTFE) and nylon 11 (PA 11) (Figure 3c). The CE−cellulose–PTFE−based TENG achieves a record−high triboelectric charge density of 533 μC m^−2^ at 95% relative humidity, four times higher than that of pure cellulose−PTFE−based TENG. This improvement is attributed to the dual−electric−polarity−augmented property of CE−cellulose, which arises from the electron−donating ether bonds and electron−withdrawing cyano−groups. The enhanced electrical output in high humidity environments, even in the presence of water droplets, paves the way for the development of TENGs for energy harvesting, self−powered sensors, wearable electronics, and human–computer interaction in high−humidity conditions. 

Hu et al. have designed a biodegradable, super−strong, and washable cellulose−based conductive macrofiber by incorporating carbon nanotubes and polypyrrole into a bacterial cellulose hydrogel [28]. This innovative macrofiber boasts high tensile strength (449 MPa), excellent electrical conductivity (5.32 S cm^−1^), and impressive stability, with minimal decreases in tensile strength and conductivity after immersion in water for a day. The macrofiber can be degraded within 108 h in a cellulase solution, highlighting its eco−friendliness (Figure 3d). The researchers created a fabric−based TENG using this cellulose−based conductive macrofiber, which produced a maximum output power of 352 μW. The fabric−based TENG can effectively charge capacitors and power commercial electronics. Furthermore, when attached to the human body, it can function as a self−powered sensor, monitoring various human motions such as walking, running, jumping, arm lifting, arm bending, and leg lifting. This study reveals the potential of biodegradable, strong, and washable conductive cellulose macrofibers in designing eco−friendly fabric−based TENGs for energy harvesting and biomechanical monitoring.

Wang et al. designed a flexible, biodegradable, single−electrode cellulose−based triboelectric nanogenerator (FR−TENG) with flame retardancy, using biocompatible black phosphorus (BP) and phytic acid (PA) as flame retardants [29]. The cellulose nanofibers (CNFs) in the system serve as a triboelectric layer, improving the performance of the TENG by increasing its contact area and charge density. The FR−TENG demonstrates a high output voltage and current at low frequency, making it suitable for wearable electronics (Figure 3e). Additionally, it exhibits a high resistance response to temperature with a thermal index of 3779.16 K in the range of 35–150 °C, enabling it to function as a temperature sensor and provide rapid fire warnings within 5 s. The CNF−BP−PA film, combined with silver nanowires (AgNWs) as a conductive layer, offers excellent fire resistance and flame retardancy. The FR−TENG’s ability to harvest energy from human motion and serve as a self−powered humidity sensor makes it ideal for multifunctional wearable electronics, temperature monitoring, and early fire warning applications in firefighting and industrial settings. 

### 3.2. Honeycomb in TENG

#### 3.2.1. Honeycomb−Structured Cellular Materials

The honeycomb structure offers numerous advantages, such as optimized space utilization, lightweight, high strength−to−weight ratio, and mechanical flexibility [42]. The research in this category explores different honeycomb−structured TENGs, such as the honeycomb−shaped TENG (H−TENG) and the 3D fabric TENG (F−TENG), for various applications such as energy harvesting from human motion, biomechanical signal monitoring, wearable power generation, and flame retardancy. The honeycomb structure’s inherent properties, including deformability and high−power density, make these TENGs promising candidates for wearable and portable electronic devices.

Yang et al. designed a honeycomb−shaped triboelectric nanogenerator (H−TENG) with a cellular structure that optimizes space utilization and power density for harvesting human motion energy and monitoring biomechanical signals [43]. The 3D−printed thermoplastic polyurethane (TPU) serves as the elastic shell, while polybutylene adipate terephthalate (PBAT) and aluminum foil act as friction layers for triboelectricity generation (Figure 4a). The adjustable honeycomb structure allows for the extension and stacking of cells based on application requirements, providing robust mechanical flexibility, high deformability, and superior fatigue resistance. H−TENGs with varying cell numbers, frequencies, accelerations, and electrode surface topographies were tested, with the maximum open−circuit voltage and power density reaching 1500 V and 10.79 W m^−2^, respectively. H−TENG can power electronic devices such as LEDs, pedometers, electronic watches, and thermometers through tapping and walking motions. Moreover, it exhibits high sensitivity in monitoring motion statuses, such as walking speed and arm bending angle. This innovative H−TENG design presents a promising solution for wearable power generation and motion monitoring devices, enhancing power density and industrial manufacturing, and potentially promoting TENG commercialization.

Tao et al. have designed a honeycomb−inspired triboelectric nanogenerator (h−TENG) that leverages the cellular structure’s compactness, light weight, deformability, and high strength−to−weight ratio for biomechanical and UAV morphing wing energy harvesting [44]. The h−TENG’s wavy surface is based on the contact triboelectrification of a cellular honeycomb structure, which enhances capacitance variation and increases output power density by dividing large hollow spaces into numerous energy generation units (Figure 4b). With excellent transparency, flexibility, and deformability, in addition to its light weight, the h−TENG is suitable for various applications, including plantar pressure mapping in shoes and flapping energy conversion in small UAV morphing wings. The article details the fabrication process, working principle, performance evaluation, and application demonstration of the h−TENG device, discussing its advantages over existing TENGs and its future development potential. The h−TENG prototype produces a peak power density of 0.275 mW cm^−3^ (or 2.48 mW g^−1^) under hand−pressing excitations, showcasing its versatility and viability for a wide range of real−world application scenarios. 

Ma et al. developed a 3D fabric triboelectric nanogenerator (F−TENG) with a honeycomb structure, made from flame−retardant yarns [45]. This cellular structure offers numerous benefits for the F−TENG, including flame retardancy, noise reduction, and enhanced energy harvesting. The yarns are created using a scalable hollow spinning fancy twister technology, which allows for the production of continuous and uniform yarns (Figure 4c). The F−TENG can serve as a self−powered escape and rescue system in fire accidents, locating survivors and guiding them to safety. Additionally, the F−TENG demonstrates excellent noise−reduction, machine washability, air permeability, durability, and repeatability features. Its potential applications include fire rescue, wearable sensors, and smart home decoration. The unique design of the honeycomb weaving structure endows the F−TENG with noise−reduction capabilities, while its flame retardancy, air permeability, and other attributes make it a promising solution for various applications in fire rescue and wearable technology, as well as smart home decoration. 

#### 3.2.2. Honeycomb−Patterned Cellular Materials 

Honeycomb−patterned surfaces provide increased contact area, better mechanical durability, and efficient charge transfer due to their unique geometric arrangement. Research in this category focuses on fabricating various materials with honeycomb−patterned surfaces, such as polymers, metals, and composites, and their integration into TENGs. These specially patterned TENGs show improved energy harvesting capabilities, higher power output, and better stability, making them suitable for applications in self−powered sensing, energy harvesting and smart textiles.

Le et al. developed a surfactant−free graphene oxide–polylactic acid (GO/PLA) nanocomposite featuring a honeycomb pattern to significantly improve the output performance of high−power antagonistic bio−triboelectric nanogenerators (A−TENG) [46]. The honeycomb pattern’s cellular structure increases the frictional area and surface charge density, thereby enhancing the TENG’s performance. The researchers utilized a two−step solution method to create customizable honeycomb patterns using suitable solvents and nonsolvents. The large surface area and electron−donating groups of GO nanoparticles greatly affected the electro−positivity and surface properties of the PLA biopolymer. The resulting A−TENG, comprising concave−honeycomb GO−PLA and convex−polydimethylsiloxane (c−PDMS), generated an output power of 3.25 mW, a 13.6−fold improvement compared to a flat−surface TENG without GO additives (Figure 5a). The use of functional GO nanoparticles without surfactants expands the potential applications of biocompatible A−TENG in healthcare, maintaining biocompatibility while enhancing output performance. 

Huang et al. devised a method for fabricating honeycomb porous microstructures (HPMs) on one−dimensional, nonplanar fiber surfaces using the breath figure (BF) technique [36]. By coating cellulose fibers with polystyrene (PS) and exposing them to humid air, the researchers enhanced the performance of silver−plated nylon fibers (SNFs) as TENGs for energy harvesting and motion sensing (Figure 5b). The HPMs improved contact electrification and charge separation efficiency by increasing surface area and roughness. Furthermore, tunable parameters allowed the researchers to regulate the electrical output and sensitivity of the SNFs. The study investigates the formation mechanism and influential factors of HPMs on nonplanar fiber substrates and demonstrates their application in SNF@HPMs−TENGs for biomechanical energy harvesting and versatile human motion sensing. The SNF@HPMs−TENG exhibited remarkable electrical performance, long−term stability, and the ability to power portable electronics and monitor various mechanical stimuli. This research provides valuable insights and guidance for regulating fiber materials’ microstructures to develop advanced functional fibers (AFFs) with customizable functionalities.

Chau et al. developed a high−performance TENG utilizing a honeycomb−patterned graphene oxide–surfactant/polylactic acid (hc−GO−S/PLA) nanocomposite film [7]. This unique structure enhances output performance by increasing the surface contact area and the surface charge density, both vital factors for TENG efficiency (Figure 5c). Fabrication of this cellular structure involves a combination of ionic interaction−supported GO dispersion and an advanced phase separation method using a chloroform and methanol solvent/nonsolvent pair. By adjusting methanol content, the honeycomb pattern’s pore diameter and density can be controlled. The TENG, comprised of hc−GO−S/PLA and its microdome−patterned polydimethylsiloxane (md−PDMS) replica, delivers an output power of 0.54 mW cm^−2^, which is 8.6 times higher than TENGs with a flat surface and no GO additives. Owing to GO’s electron−donating properties, excellent biocompatibility, and strong interaction with PLA, the hc−GO−S/PLA nanocomposite has potential applications in biotechnology, optics, and catalyst fields. The honeycomb−patterned surface not only boosts surface charge density but also enlarges the frictional surface area of the GO/PLA nanocomposite, making it a promising component for high−performance TENGs.

Bui et al. developed a TENG featuring a honeycomb−patterned polyimide (hc−PI) film, addressing the limitations of traditional TENGs in terms of durability and resistance to harsh conditions [47]. The honeycomb structure optimizes electrification efficiency by increasing contact area and creating air gaps between the triboelectric materials. The hc−PI film is fabricated using a scalable and improved phase separation method using methanol as a pattern inducer and droplet template stabilizer (Figure 5d). The TENG, composed of hc−PI as the negative triboelectric material and copper foil as the positive material and electrode, demonstrates an output power of 1.05 W m^−2^, 22 times higher than a TENG with a flat PI film. It also exhibits excellent durability, withstanding 20,000 contact–separation cycles, and functions stably at temperatures up to 200 °C. Compared to commercial Kapton and self−made flat PI film TENGs, the hc−PI TENG offers superior electrical output, flexibility, thermostability, and durability, making it ideal for self−powered electronics and sensors in harsh environments. The innovative fabrication method and honeycomb structure of the hc−PI film contribute to the enhanced performance of this TENG, expanding its potential for practical applications in various industries. 

## 4. Cellular Materials in TENG−SC

### 4.1. Energy Harvesting and Storage

#### 4.1.1. Cellular Structure in TENG−SC 

In recent studies, cellular structures have been shown to enhance energy collection and storage in TENGs and supercapacitors (SCs). These innovative designs improve the performance and flexibility of energy conversion and storage systems by increasing surface area and contact area, thus enabling more efficient energy harvesting for powering electronic devices.

Jayababu et al. have designed an integrated energy conversion and storage system comprising flexible ZnO nanorods (NRs) and a conductive carbon black (CB) nanocomposite (NC) −based triboelectric nanogenerator (FZCT) and supercapacitor (FZCS) (Figure 6a) [48]. The honeycomb−like structure of this system enhances the performance and flexibility of both the TENG and SC by increasing their surface area and contact area. With optimized CB content, the FZCT delivers improved electrical output, while the FZCS exhibits high areal capacitance, energy density, and power density, as well as excellent cycling stability. The supercapacitor in this integrated device plays a crucial role in storing the electrical energy generated by the TENG, providing a stable power supply for wearable electronics. The system efficiently harvests ambient mechanical energy and stores it for powering various electronic devices, such as LEDs, digital watches, and thermometers. The use of a straightforward precipitation method for fabricating the flexible honeycomb−like structure contributes to the overall effectiveness and adaptability of this integrated energy conversion and storage system. 

In a related development, Dong et al. have introduced an innovative supercapacitor−inspired triboelectric nanogenerator (SI−TENG) that utilizes an electrostatic double layer, akin to supercapacitors (Figure 6b) [49]. The device features 3D electrodes, made from polyurethane sponge and silver nanowires, that adopt a cellular, honeycomb−like structure. This design enhances the surface area for charge adsorption, storage, and flexibility while providing compressibility for mechanical energy harvesting. The novel TENG demonstrates high output performance and long−term stability, achieved by employing electrochemical polarization charging or corona charging for high surface charge density. 

The SI−TENG, inspired by supercapacitor principles, can be integrated with an actual supercapacitor to create a self−charging power system suitable for flexible electronics. Dong et al.’s findings reveal a proof−of−concept TENG based on an electrostatic double layer, opening up new opportunities for energy conversion and advancement in the field of chemical principles. The integration of these energy harvesting and storage technologies showcases the potential for sustainable and efficient solutions in powering wearable electronics and portable devices.

#### 4.1.2. Flexible and Stretchable Structure in TENG−SC

Flexible and stretchable structures in TENG−SC systems have become increasingly important in the development of wearable electronics and portable devices, as they contribute to mechanical durability, adaptability, and user comfort [53,54,55,56,57,58,59,60,61]. Yang et al. focus on creating a fully stretchable self−charging power unit (FS−SCPU) by integrating a micro−supercapacitor (MSC) and a TENG using oxidized single−walled carbon nanotube (Ox−SWCNT)/polymer electrodes [50]. The honeycomb−like cellular structure provides mechanical durability and flexibility, allowing the energy devices to withstand tensile and compressive loads while conforming to different shapes and surface curvatures. This adaptability and the auxetic properties of the structure enhance its stretchability and energy dissipation, contributing to the overall efficiency and durability of the TENG−SC systems.

MSCs play a vital role in storing the electricity generated by TENGs, which is then used to power wearable electronics. Offering high power density and exceptional cycle life, MSCs are well−suited for on−chip wearable devices (Figure 6c). The FS−SCPU demonstrates high capacitance, mechanical flexibility, and stretchability over 10,000 stretching test cycles. It also showcases its ability to charge the MSC from 0 to 2.2 V in 1200 s and power a commercial digital clock for approximately 10 s, highlighting the potential of stretchable polymer composites with O_x_−SWCNTs as optimal electrodes and active materials for fully stretchable, self−powered wearable electronics. 

In another study, Hu et al. design a tough, triple−network organohydrogel for wearable intelligent devices, capitalizing on its unique porous honeycomb structure for enhanced mechanical strength and water absorption [51]. The organohydrogel comprises poly (vinyl alcohol), sodium alginate, cellulose nanofibrils (CNFs), MXene–graphene oxide (MX–GO) nanocomposites, and ethylene glycol (Figure 6d). The honeycomb structure of CNFs contributes to the organohydrogel’s high mechanical properties, water retention capacity, and adaptability to different shapes and curvatures. The MX–GO nanocomposites within the organohydrogel improve its capacitive charge storage ability and triboelectric output, facilitating its application in supercapacitors and TENGs. The organohydrogel−based supercapacitor exhibits a high specific capacitance of 5.4 F g^−1^ and excellent cycling stability, paving the way for the development of next−generation wearable intelligent devices. 

Together, these examples highlight the importance and potential of flexible and stretchable structures in TENG−SC systems for wearable electronics and portable devices, demonstrating their contributions to mechanical durability, adaptability, and user comfort.

#### 4.1.3. Textiles in TENG−SC

Textiles in TENG−SC systems have emerged as a promising area of research, offering innovative solutions for wearable electronics and portable devices [62,63,64,65,66,67,68,69,70,71,72,73,74]. Ren et al. have successfully designed a self−charging power cloth that combines a knitted TENG textile with asymmetric all−solid−state supercapacitor yarns [75]. The TENG textile, embedded in the fabric, captures biomechanical energy from human motion, while the supercapacitor yarns store the generated electricity. Notably, the knitting technology used in the production of these textiles results in a cellular structure that provides high stretchability, elasticity, and flexibility, making the materials ideal for wearable applications. Supercapacitors play a crucial role in this system by storing the electrical energy harvested by the TENG textile, thereby providing a stable power supply for wearable electronic devices. Characterized by high power densities, long cycle lives, rapid charging/discharging rates, and eco−friendliness, asymmetric all−solid−state supercapacitor yarns utilize Co_3_O_4_ nanosheet arrays on carbon fiber bundles as the positive electrode material and active carbon as the negative electrode material. The self−charging power cloth showcases exceptional elasticity, flexibility, stretchability, breathability, affordability, and compatibility with traditional textile processing. This innovative design can harvest and store biomechanical energy from human motion, offering a sustainable energy source for wearable electronics and portable electronic devices. 

Similarly, Park et al. have developed a novel hybrid energy system that integrates a TENG and a supercapacitor (SC) using electrospun nanofiber and honeycomb−structured wrinkled polystyrene (PS) as electrodes (Figure 6e) [52]. The unique cellular structure of the wrinkled PS substrate not only enhances the effective surface and contact areas of the electrodes but also contributes to the improvement in the TENG’s surface charge density and the electroactive area of the SC. Much like in Ren et al.’s work, supercapacitors serve a pivotal function in this system by storing the electrical energy generated by the TENG and supplying a stable power source for small electronic devices. These devices boast superior power densities, rapid charge/discharge rates, extensive cycle life, and lower maintenance costs compared to batteries. Polyaniline (PANI) is employed as an electroactive material for the supercapacitor due to its high conductivity, stability, and affordability. Ultimately, the innovative system efficiently channels electrical energy from TENG into the supercapacitor, successfully powering a commercial green LED with the stored energy. 

In both cases, supercapacitors play a critical role in textile−based TENG−SC systems by storing the electrical energy generated by the TENG and providing a stable power supply for electronic devices. The textile−based designs offer numerous advantages such as high−power density, rapid charging/discharging rates, extensive cycle life, and eco−friendliness. The incorporation of these textile technologies in TENG−SC systems demonstrates the potential for sustainable energy solutions that can be seamlessly integrated into wearable electronics and portable devices.

### 4.2. Bioelectronics and Multi−Functional Applications

#### 4.2.1. Bioelectronics Based on TENG−SC

Nerve Stimulation with Soft, Substrate−free Organic Bioelectronics: Strakosas et al. present an innovative method for creating soft, substrate−free organic bioelectronics within living systems by leveraging endogenous metabolites to initiate enzymatic polymerization of organic precursors [76]. The cellulose structure serves as a scaffold for developing conducting polymers within the gel matrix, utilizing cellulose nanofibrils (CNF) to create a 3D network with enhanced electrical properties. The incorporation of supercapacitors based on polypyrrole (PPy)−coated CNF electrodes demonstrates the potential of this approach for nerve stimulation, as the supercapacitors can store electrical energy and deliver it to target tissues. This groundbreaking approach addresses the incompatibility between conventional rigid bioelectronics and living systems, enabling the development of fully integrated, in vivo−fabricated electronics within the nervous system for therapeutic purposes.

Long et al. developed a biomass−mediated strategy to synthesize nitrogen−doped carbon aerogels (C−NGD), employing a calcinated mixture of glucose and dicyandiamide nanosheets (C−GD) and cellulose nanofibers (CNFs) as sustainable, affordable, and abundant precursors (Figure 7a) [77]. The cellular structure of these carbon aerogels bolsters mechanical strength, porosity, surface area, and charge transport, while enabling ion diffusion pathways. These structural attributes also impact the material’s electromagnetic wave absorption capabilities. Supercapacitors, which are renowned for their high power density, fast charge/discharge rates, long cycle life, and eco−friendliness, benefit from the unique features of carbon aerogels, including high conductivity, large surface area, and tunable pore structures. The C−NGD demonstrates exceptional elasticity, compressibility, fatigue resistance, and a super−stable wave−layered structure, supporting ultra−high compression strains and long−term compression cycles. Additionally, its wide−range linear sensitivity under a 0–10 kPa pressure range showcases its potential for wearable piezoresistive sensing devices, capable of detecting body motion and bio−signals. Furthermore, the versatility of C−NGD lends itself to promising applications in supercapacitors and TENGs, solidifying its relevance in flexible electronics and energy conversion/storage devices. 

Edible Energy Harvesting Devices with Ethyl Cellulose Composites: Lamanna et al. explore the use of ethyl cellulose and activated carbon composites for edible energy harvesting devices, focusing on their application in TENGs and SCs (Figure 7b) [78]. The biodegradable and edible cellulose structure ensures flexibility, solubility, and durability of the composites, making them suitable for integration into wearable devices that harvest energy from daily activities, such as walking or running. The successful implementation of these composites as electropositive elements in TENGs and electrodes in SCs highlights their potential for high voltage outputs, power densities, energy densities, and cycling stability, paving the way for new opportunities in environmentally friendly and edible energy harvesting devices.

In summary, these three examples illustrate the significant advancements in the field of biologically related TENGs, focusing on their application in self−powered biomedical sensors, edible energy harvesting devices, and nerve stimulation. The exploration of novel materials such as nitrogen−doped carbon aerogels, ethyl cellulose composites, and soft organic bioelectronics within living systems highlights the potential for innovative applications in healthcare, wearable technology, and therapeutic interventions.

#### 4.2.2. Multi−Functional TENG−SCs That Are Based on Cellular Materials

In the rapidly evolving field of TENGs and supercapacitors (TENG−SC), multifunctionality has become a crucial aspect for the development of next−generation energy devices [80,81,82,83]. Multifunctional TENGs are capable of performing multiple tasks or having multiple functions, rendering them highly versatile and useful across a wide range of applications. This multifunctionality allows TENG−SC devices to not only harvest energy from various mechanical sources but also to serve as sensors, actuators, self−powered systems, and wearable technologies. Therefore, the integration of multifunctional TENGs into energy devices holds great potential to revolutionize various fields, from healthcare to environmental monitoring and beyond.

Hu et al. introduce a multifunctional device that synergistically combines a TENG with a structural supercapacitor, utilizing woven carbon fiber (WCF) electrodes reinforced by P−doped Cu−Mn selenide nanowires (Figure 7c) [10]. Central to this design is the honeycomb−like cellular structure, which imparts stretchable and flexible qualities to the electrode materials. This structure also enhances the supercapacitor’s specific capacitance and energy density by providing more pores for ion transport. Acting as an energy storage medium, the supercapacitor stores the electrical energy harvested by the TENG, boasting high power densities, fast charge/discharge rates, and long cycle lives compared to batteries. Exhibiting remarkable mechanical and electrochemical properties, the integrated device promises to meet future energy demands in self−charging automobiles, electronics, and a variety of outdoor applications. 

Building upon the multifunctional concept, Guo et al. put forward a self−charging electrochromic supercapacitor device (SC−ESCD) that seamlessly integrates a sliding−mode direct−current triboelectric nanogenerator (DC−TENG) with an electrochromic supercapacitor (ESCD) (Figure 7d) [79]. The SC−ESCD is capable of converting mechanical sliding energy into electricity, storing it within the ESCD, and producing optical responses to sliding motions. The ESCD incorporates polyaniline (PANI) as the active material and features a 3D−printed honeycomb structure, which augments its mechanical stability, flexibility, and electrochemical performance. Exhibiting high areal capacitance and stable cycling performance, the versatile SC−ESCD can be customized into various patterns to suit diverse applications. This cutting−edge device functions as both an optically responsive component reacting to sliding motion stimuli and a self−charging power source for other electronics, holding immense potential for next−generation smart electronics such as smart windows, information displays, and self−powered devices.

Taking this concept further, Zhang et al. develop a high−efficiency self−charging power system (SCPS) specifically designed for outdoor search and rescue applications by integrating a hybrid nanogenerator (HNG) and an asymmetric supercapacitor (ASC) (Figure 7e) [22]. The HNG, which consists of a rotational triboelectric nanogenerator (R−TENG) and an electromagnetic generator (EMG), proficiently harvests electrical energy from rotary motion. Honeycomb−like NiCo−layered double hydroxide (LDH) nanosheets function as the ASC’s electrodes, affording high energy density and rapid charge/discharge rates. This porous structure bolsters the electrochemical performance of the ASC by presenting more active sites for ion transfer and charge storage. The SCPS is devised to power a custom−made positioning device for outdoor search and rescue using environmental mechanical energy. The paper details the design, fabrication, characterization, and performance evaluation of the SCPS components and system. The SCPS can achieve 5.9 V after 120 s at a rotation speed of 200 rpm, illustrating its practicality for real−world applications. The R−HNG’s output is enhanced by modifying the work function difference between the friction layer and electrode, with the working principle elucidated using Kelvin probe force microscopy (KPFM) and finite element simulation. By tackling the long charging time issue, this work provides a promising solution for constructing high−performance SCPSs and advancing their development in the Internet of Things (IoT) domain.

Through the integration of TENGs with supercapacitors and the incorporation of innovative structural designs, these devices have made significant advancements in energy harvesting, storage, and utilization. These developments have resulted in devices that are highly efficient, reliable, and adaptable to a wide range of application scenarios.

### 4.3. The Role of Cellular Materials in Enhancing the Performance of TENG−SC Systems 

Cellular materials, such as porous carbon, have been shown to enhance the performance of TENG−SC (supercapacitor) systems by providing a higher surface area for charge storage and efficient charge transfer. The porous structure of cellular materials provides a large surface area−to−volume ratio, which allows for more active sites for charge storage and improved ion transport. This leads to increased energy storage capacity and faster charge/discharge rates.

In TENG−SC systems, the TENG converts mechanical energy into electrical energy, which is stored in the supercapacitor. By incorporating cellular materials into the supercapacitor electrode, the energy storage capacity of the system can be significantly improved. Additionally, the porous structure of the cellular materials can also improve the performance of the TENG by increasing the effective contact area between the TENG and the counter electrode, leading to higher output voltages and currents.

In summary, cellular materials can enhance the performance of TENG−SC systems by providing a high surface area for charge storage and efficient charge transfer. The use of cellular materials in TENG−SC systems has the potential to improve energy generation and storage, which can be useful in a variety of applications such as energy harvesting and storage, wearable electronics, and biomedical devices.

There are differences in output between different TENGs and SCs. Table 1 provides a summary of the parameters of TENG or SC cell materials in recent years (Table 1).

There are also some new materials for supercapacitors that have been reported and confirmed to have practical value [86,87,88], demonstrating their potential in TENG−SCs, and people can choose among them according to their actual needs.

## 5. Conclusions and Perspectives

### 5.1. TENGs That Are Based on Cellular Materials

#### 5.1.1. Enhancing Power Output

To improve the power output of TENGs in the future, researchers can investigate methods such as enhancing surface charge density, optimizing contact area, and using cellular materials with better triboelectric properties. This may involve exploring various surface treatments and nanoscale surface patterning techniques to maximize charge transfer and energy conversion efficiency. Researchers can study the influence of mechanical force (e.g., frequency, amplitude, and mode) on TENG output and develop strategies to maximize performance under different force conditions. Understanding the interplay between these factors can lead to the design of adaptive TENG systems that can efficiently harvest energy from diverse mechanical sources. Researchers can also optimize device design, including electrode configuration and geometric shapes, to boost output power and efficiency. Investigating novel geometries, multilayer structures, and hybrid materials can potentially lead to significant improvements in power generation capabilities.

#### 5.1.2. Biocompatibility and Comfort

In the future, researchers can focus on improving the biocompatibility and comfort of TENGs by researching biocompatible, non−toxic, and skin−friendly materials, particularly when designing wearable or implantable devices. Ensuring the safety and comfort of users is crucial in promoting the adoption of TENG−based wearables in various applications, from healthcare to fitness monitoring. Researchers can investigate flexible, stretchable, and lightweight cellular materials and cellular designs that can comfortably conform to the human body or other irregular surfaces. This may involve exploring the use of polymers, textiles, and other soft materials in combination with TENG structures to create devices that are both functional and comfortable to wear.

#### 5.1.3. Wireless Communication

To enhance wireless communication capabilities, researchers can develop TENG−based energy harvesting systems for powering wireless communication modules in IoT devices, wearables, or sensor networks. This will enable self−powered systems that can operate for extended periods without the need for battery replacements or external charging, paving the way for more sustainable and low−maintenance electronic devices. Researchers can investigate the integration of TENGs with wireless power transmission technologies to enable remote energy transfer and reduce the need for physical connections. This can lead to innovative applications in remote sensing, environmental monitoring, and smart infrastructure, where access to power sources is often limited.

#### 5.1.4. Wearability and Durability

In order to improve wearability and durability of TENG devices that are based on cellular materials or designs, researchers can focus on designing TENG devices that are not only flexible and comfortable but also resistant to wear, tear, and environmental factors (e.g., moisture, temperature, UV radiation). This may involve the development of protective coatings, encapsulation techniques, and advanced materials with inherent resilience to harsh conditions. Researchers can study the long−term performance and degradation of TENG materials and structures and develop strategies for improving their durability and maintenance. This can include investigating the effects of mechanical fatigue, chemical exposure, and environmental stress on TENG performance and identifying methods to mitigate these factors, ensuring the longevity and reliability of TENG−based devices.

### 5.2. Supercapacitors That Are Based on Cellular Materials

#### 5.2.1. Increasing Energy Density

In the future, researchers can focus on developing novel cellular electrode materials with a high energy density to improve the energy storage capacity of supercapacitors. They can explore nanomaterials, hybrid materials, and conductive polymers that have cellular structures and can store more charge per unit volume. Additionally, researchers can work on optimizing the balance between energy density and power density, ensuring that supercapacitors can deliver high power while storing more energy.

#### 5.2.2. Improving Stability and Reliability

To enhance the long−term stability and reliability of cellular supercapacitors, researchers can investigate protective coatings or surface treatments for electrode materials. This could help prevent degradation due to environmental factors, such as temperature and humidity. Furthermore, they can develop advanced electrolytes with higher thermal stability and better compatibility with cellular electrode materials, which would lead to improved cycle life and overall performance.

#### 5.2.3. Enhancing Charge and Discharge Rates

Researchers can investigate new methods to improve charge transfer at the electrode−electrolyte interface, potentially using novel catalysts or functional groups that facilitate faster ion exchange. Additionally, they can explore the 3D architectures and nanostructures of electrode materials to increase the available cellular surface area and improve ion transport pathways. This would result in enhanced charge and discharge rates, making supercapacitors developed from cellular materials more attractive for high−power applications.

#### 5.2.4. Exploring New Applications

In the future, researchers can work on developing flexible and miniaturized supercapacitors from cellular materials and structures, which can be integrated into wearable devices and medical implants. These supercapacitors could provide the necessary power for sensors, data transmission, and on−demand drug delivery. Furthermore, they can investigate the potential of supercapacitors for grid−scale energy storage, working in conjunction with renewable energy sources to provide stable power to the grid during peak demand periods.

#### 5.2.5. Cost Reduction

To make cellular supercapacitors more affordable, researchers can focus on developing low−cost, scalable fabrication methods and materials. They can investigate alternative precursor materials or synthesis routes that lead to high−performance cellular electrode materials at a lower cost. Additionally, researchers can explore eco−friendly, low−cost electrolytes that offer good performance without the need for expensive, toxic, or environmentally harmful components. By optimizing the manufacturing process and implementing recycling and reuse strategies, the overall cost of supercapacitors can be reduced, making them more accessible for widespread use.

### 5.3. TENG−SC

#### 5.3.1. Material Design

Various strategies can be employed to enhance the performance of TENG−SC through material design. One method involves creating cellular materials with high triboelectric charges, increasing the energy generation of the TENG component. This can be achieved by modifying the materials’ surface chemistry, morphology, or composition. Another approach is designing cellular materials with high capacitance, which boosts the energy storage capacity of the supercapacitor component. This can be accomplished by utilizing cellular materials with a high specific surface area, such as porous carbons or metal oxides. Future researchers can investigate developing novel materials or modifying existing ones to augment triboelectric and capacitive properties, leading to improved TENG−SC performance.

#### 5.3.2. Structural Design

Optimizing TENG−SCs’ structural design can also enhance performance. One method is designing the cellular electrode structure to increase surface area for greater charge storage, which can be achieved using cellular materials with a hierarchical pore structure or by designing the electrode in a 3D architecture. Another approach is modifying the cellular separator material to reduce the supercapacitor’s internal resistance, improving energy density and power output. Researchers can focus on creating innovative structural designs and novel cellular separator materials to boost overall efficiency and energy density in TENG−SC devices in the future.

#### 5.3.3. Device Integration

Integrating TENG−SC with other devices can lead to performance improvements. For instance, combining the TENG component with a solar cell can expand the system’s energy harvesting capacity. The TENG can harvest energy from mechanical motion, while the solar cell collects energy from light. The supercapacitor can then store energy from both sources for later use, enhancing the overall energy efficiency of the system. In the future, researchers can explore new methods of integrating TENG−SC with diverse energy harvesting devices, such as piezoelectric, thermoelectric, or other renewable energy technologies, to further amplify energy generation and storage capabilities within the entire system.

#### 5.3.4. System Optimization

Optimizing the entire TENG−SC system is crucial for enhancing overall performance. This process entails improving energy conversion efficiency, minimizing losses, and increasing energy density. One approach is optimizing control circuitry, which can enhance energy conversion efficiency and reduce losses. Another method is optimizing electrode and separator materials to increase the supercapacitor component’s energy density. Furthermore, optimizing device integration can boost the system’s overall energy harvesting capacity, resulting in better performance. Researchers can concentrate on developing advanced control algorithms and innovative device integration strategies, as well as optimizing materials, in order to achieve a more efficient and high−performance TENG−SC system in the future. 

### 5.4. Future Application Directions

As the demand for sustainable and efficient energy harvesting and storage solutions continues to grow, there is great potential for the further development and integration of cellular materials in TENG−SC systems. Some possible future directions for this field include:

Integration with emerging materials: TENG−SC systems can be further improved by integrating them with emerging materials such as graphene, MXenes, and metal–organic frameworks. These materials offer unique properties such as high conductivities, tunable porosities, and high surface areas, which can enhance the performance and efficiency of TENG−SC systems.

Development of flexible and stretchable cellular materials: As wearable and implantable devices become more prevalent, the development of flexible and stretchable cellular materials becomes essential. These materials can provide both mechanical compliance and efficient energy storage, enabling the development of self−powered wearable and implantable devices.

Multi−functional cellular materials: Cellular materials can be designed to be multi−functional, enabling the simultaneous harvesting, storage, and sensing of mechanical energy. This can greatly simplify the design of TENG−SC systems, reducing their complexity and cost.

Scalable fabrication of cellular materials: The scalability of the fabrication of cellular materials is crucial for their commercialization and widespread use. Advances in scalable fabrication methods, such as 3D printing and roll−to−roll processing, can enable the efficient and cost−effective production of cellular materials for TENG−SC systems.

Real−world applications: The development of TENG−SC systems for real−world applications, such as powering sensors, wearables, and IoT devices, will require significant advances in their efficiency, durability, and reliability. Continued research in this field is necessary to realize the full potential of TENG−SC systems for practical use.

In conclusion, cellular materials have emerged as essential components in TENG−SC systems, enabling improved performance and efficiency of sustainable energy harvesting and storage solutions. Their unique structural characteristics such as high surface−to−volume ratios, mechanical compliance, and customizable properties make them ideal for a wide range of applications. As the field of TENG−SC systems continues to develop, the integration of cellular materials holds great promise for the realization of efficient, sustainable, and scalable energy harvesting and storage solutions.

## Figures and Tables

**Figure 1 materials-16-03751-f001:**
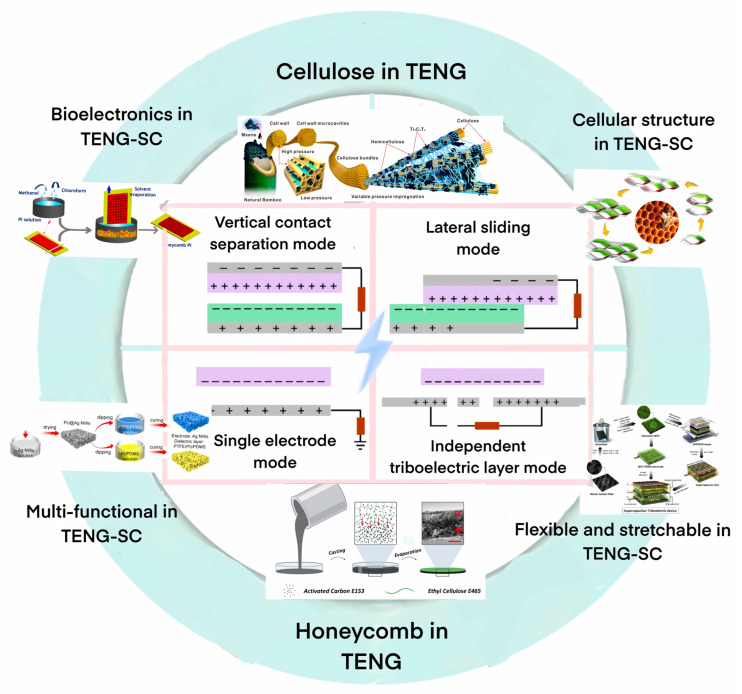
Cellular Materials in TENG−SCs.

**Figure 2 materials-16-03751-f002:**
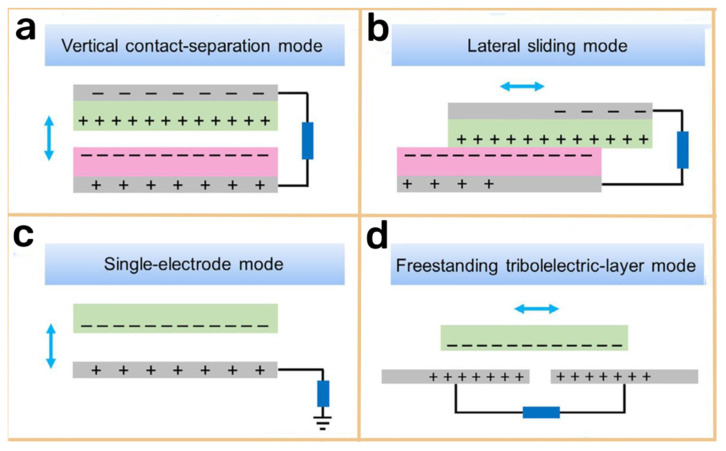
The following four modes are commonly used in the field of TENG: (**a**) Vertical Contact2Separation Mode, (**b**) Lateral Sliding Mode, (**c**) Single−Electrode Mode, (**d**) Freestanding Triboelectric−Layer Mode. Reproduced with permission from [20] (Reprinted/adapted with permission from Ref. [20]. 2022, MDPI).

**Figure 3 materials-16-03751-f003:**
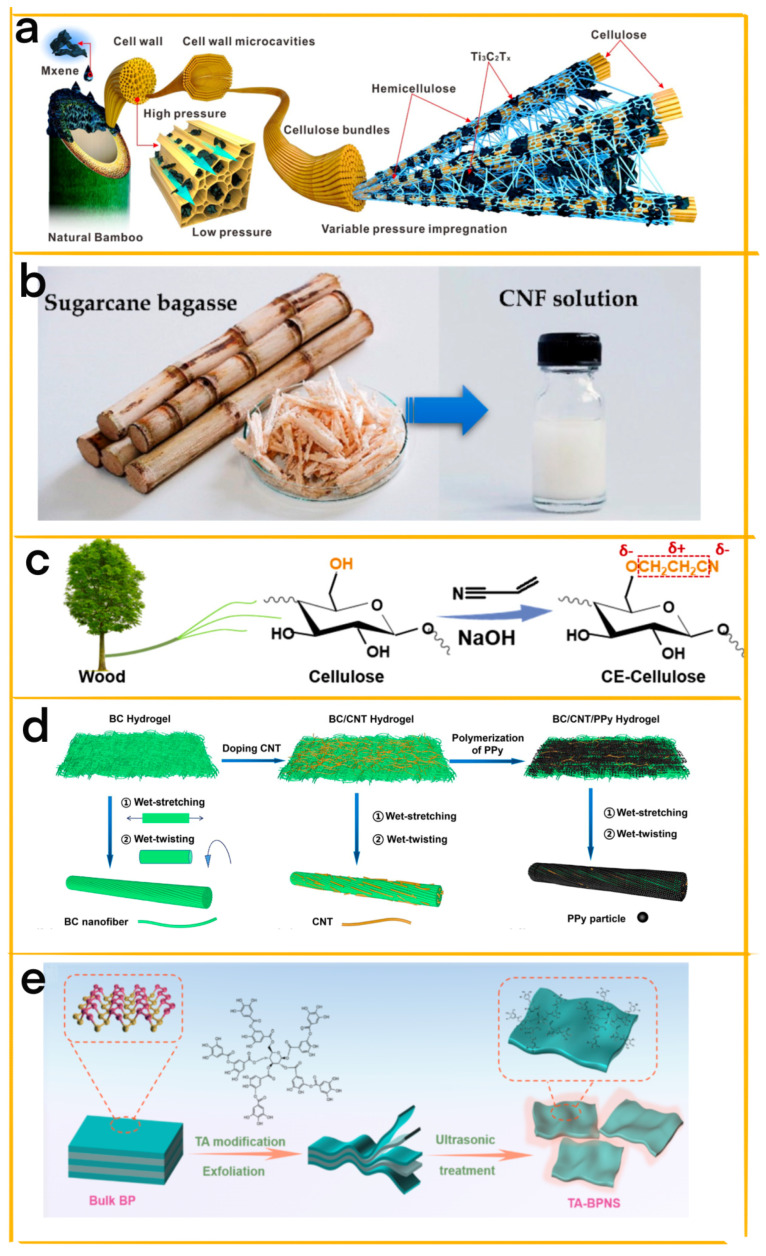
(**a**) Schematic diagram illustrating the preparation and structural connection of the cellulose/Ti_3_C_2_T_x_ composite [25] (Reprinted/adapted with permission from Ref. [25]. 2023, Elsevier). (**b**) Preparation of CNF solution from sugarcane bagasse for the fabrication of NR−CNF composite films [26] (Reprinted/adapted with permission from Ref. [26]. 2022, Elsevier). (**c**) Synthesis process of CE−cellulose depicted [27] (Reprinted/adapted with permission from Ref. [27]. 2022, Elsevier). (**d**) Schematic representation of BC, BC/CNT, and BC/CNT/PPy macrofiber fabrication [28] (Reprinted/adapted with permission from Ref. [28]. 2022, Springer Nature). (**e**) Schematic outline of the preparation of TA−BPNS [29] (Reprinted/adapted with permission from Ref. [29]. 2022, Elsevier).

**Figure 4 materials-16-03751-f004:**
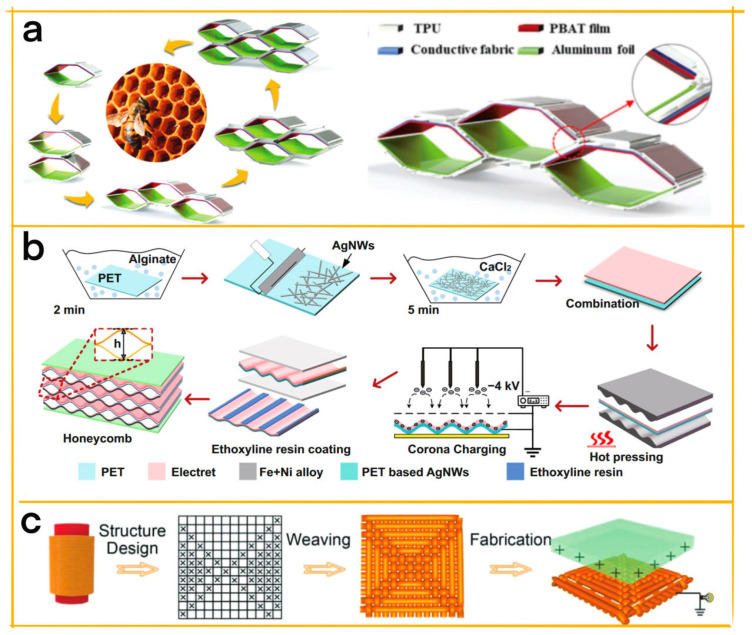
(**a**) Illustration of the honeycomb−shaped extension method for H−TENG [43] (Reprinted/adapted with permission from Ref. [43]. 2022, Wiley−Blackwell). (**b**) Fabrication processes and working mechanisms of the proposed h−TENG [44] (Reprinted/adapted with permission from Ref. [44]. 2021, Springer Nature). (**c**) Schematic of the fabrication process for the flexible 3D F−TENG [45] (Reprinted/adapted with permission from Ref. [45]. 2020, John Wiley and Sons).

**Figure 5 materials-16-03751-f005:**
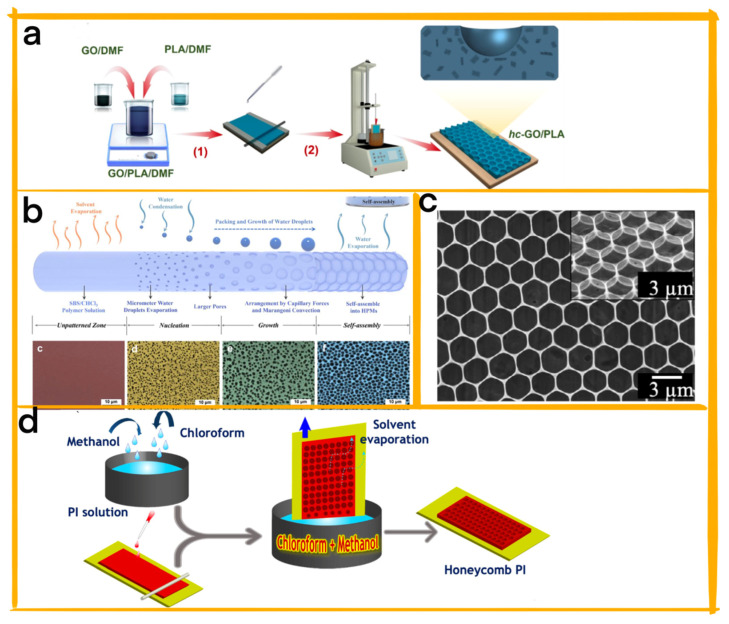
(**a**) Schematic illustration of the hc−GO/PLA film fabrication process using an enhanced phase separation method [46] ( Reprinted/adapted with permission from Ref. [46]. 2022, VIETNAM NATL UNIV). (**b**) Depiction of the various stages of the BF method applied to 1D fiber [36] (Reprinted/adapted with permission from Ref. [36].2022, Elsevier). (**c**) Low−and high−magnification SEM images of the hc−GO−S/PLA film. The inset of (**c**) showcases the tilt−view SEM image of the honeycomb film [7] (Reprinted/adapted with permission from Ref. [7]. 2021, Wiley). (**d**) Schematic representation of the hc−PI film fabrication process utilizing scalable solution coating techniques [47] (Reprinted/adapted with permission from Ref. [47]. 2022, Amer Chemical Soc).

**Figure 6 materials-16-03751-f006:**
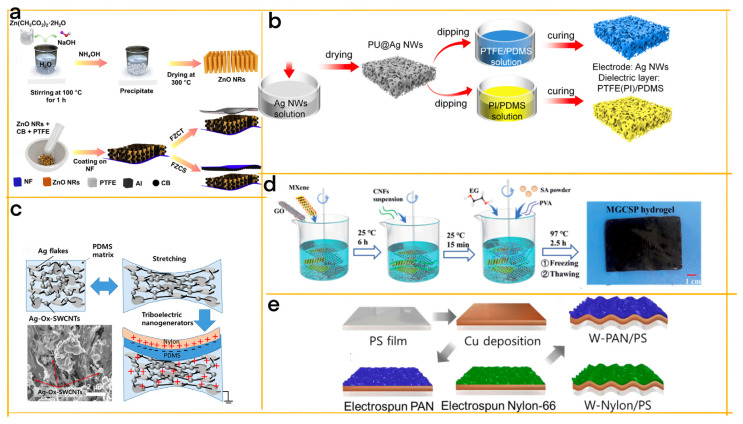
(**a**) ZnO NRs synthesis and FZCT and FZCS fabrication [48] (Reprinted/adapted with permission from Ref. [48]. 2021, Elsevier). (**b**) Schematic diagram of the 3D electrode pair preparation for SI−TENG [49] (Reprinted/adapted with permission from Ref. [49]. 2022, Elsevier). (**c**) Schematic and SEM image of the stretchable Ag−Ox−SWCNT/Ag flake/PDMS current collector for MSC and FTENG [50] (Reprinted/adapted with permission from Ref. [50].2021, Elsevier). (**d**) Illustration of the MGCSP organohydrogel preparation process [51] (Reprinted/adapted with permission from Ref. [51]. 2022, Royal Soc Chemistry). (**e**) Fabrication of W−PAN/PS and W−Nylon/PS electrodes [52] (Reprinted/adapted with permission from Ref. [52]. 2022, MDPI).

**Figure 7 materials-16-03751-f007:**
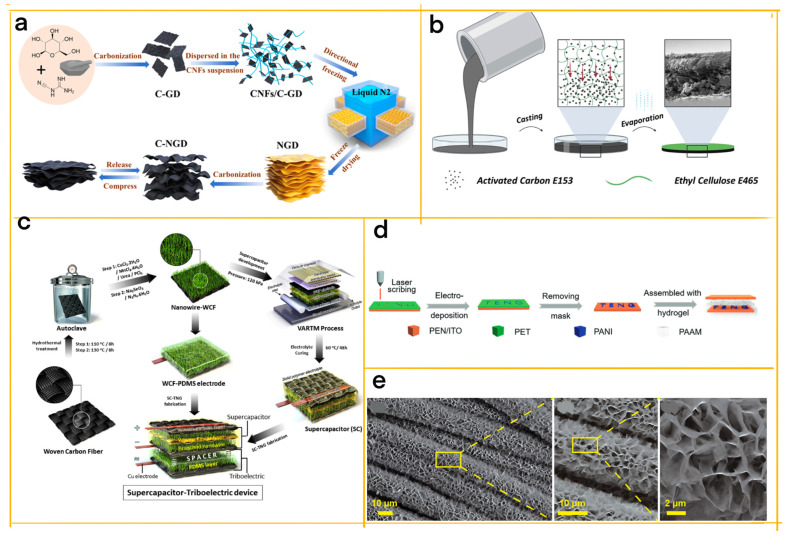
(**a**) Schematic illustration of the C−NGD preparation via a multi−step method [77] (Reprinted/adapted with permission from Ref. [77]. 2021, Elsevier). (**b**) Edible positive triboelectric film manufacturing process: ethanol−AC−EC slurry is drop−cast in a Petri dish and left to evaporate, forming AC precipitates and a layered biphasic film. The inset shows an SEM cross−section image of a 40% AC sample (scale bar: 35 μm) [78] (Reprinted/adapted with permission from Ref. [78]. 2023, Elsevier). (**c**) Schematic presentation of P@Cu–Mn selenide nanowires on WCF, followed by supercapacitor development using the VARTM process and fabrication of the triboelectric and supercapacitor device [10] (Reprinted/adapted with permission from Ref. [10]. 2020, Elsevier). (**d**) Schematic illustration of the fabrication process [79] (Reprinted/adapted with permission from Ref. [79]. 2021, Wiley−V C H Verlag Gmbh). (**e**) SEM images of NiCo−LDH at various magnifications [22] (Reprinted/adapted with permission from Ref. [22]. 2022, Elsevier).

**Table 1 materials-16-03751-t001:** Summary of parameters of cellular materials TENGs or SCs.

Date	Materials	Outputs	Device	Working Modes	Reference
2020	P@Cu_0.5_Mn_0.5_Se_2_−PDMS@WCF	7.4 W m^−2^	SCs	Contact–separation	[10]
2021	ZnO NRs@CB NCs	27.44 mW cm^−2^	SCs	Contact–separation	[84]
2021	FEP−AgNWs	0.275 mW m^−3^	TENGs	Honeycomb	[85]
2021	PEN/ITO−PET−PANI−PAAM	15.2 mA cm^−2^	SCs	Contact–separation	[79]
2022	HNG−NiCo@LDH	749.6 W kg^−1^	SCs	Freestanding	[22]
2022	PBAT−conductive fabric	10.79 W m^−2^	TENGs	Contact–separation	[43]

## Data Availability

Data available on request from the authors.

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
