# Peer review of "The Integration of Triboelectric Nanogenerators and Supercapacitors: The Key Role of Cellular Materials"

_materials, 2023, doi:10.3390/ma16103751_

Round 1

Reviewer 1 Report

1. Authors should revise with the different proper key results including numerical hints in the abstract and conclusion part since it's more identical.

2. Express the units used in the manuscript uniformly. For example, use either ml or mL. Please keep the same style for Figure.* or Fig.* throughout the main text. Use h instead of hour/s,

3. Some typo errors have been obtained in this manuscript. The manuscript needs extensive editing and the quality of the paper improved substantially.

4. An author should compare the results with a table in a recent report for your novel outcomes.

5. The author should update the latest works, some important and recent works are missing from the reference section.

Minor editing of the English language required

Author Response

Dear reviewer:

Re: Manuscript ID: materials-2352249 and Title: The Integration of Triboelectric Nanogenerator and Supercapacitors: The Key Role of Cellular Materials

Thank you for your comments concerning our manuscript entitled “The Integration of Triboelectric Nanogenerator and Supercapacitors: The Key Role of Cellular Materials” (ID: materials-2352249). Those comments are valuable and very helpful. We have read through comments carefully and have made corrections. Based on the instructions provided in your letter, we will upload the revised manuscript.

The specific modifications are as follows:

Q1. Authors should revise with the different proper key results including numerical hints in the abstract and conclusion part since it's more identical.

Modification: Tables and numbers have been added to the text to make the research results more intuitive. The abstract and conclusion have been revised to highlight the work and significance of this article.

Q2. Express the units used in the manuscript uniformly.

Modification: The use of units is unified in the text. For example, Wcm-2 instead of W /cm2, etc.

Q3. Some typo errors have been obtained in this manuscript. The manuscript needs extensive editing and the quality of the paper improved substantially.

Modification: Checked and polished the English words, grammar, and sentence structures in the article to make it more coherent and articulate.

Q4. An author should compare the results with a table in a recent report for your novel outcomes.

Modification: A table has been added to the article to summarize the research status of honeycomb structures of TENGs and SCs in recent years, including date, output, etc.

Q5. The author should update the latest works, some important and recent works are missing from the reference section.

Modification: The article has added references from this year (2023) to improve the reference content.

We would love to thank you for allowing us to resubmit a revised copy of the manuscript and we highly appreciate your time and consideration.

Sincerely.

Reviewer 2 Report

The author explored the recent advancements in integrating TENGs and supercapacitors with cellular materials. They discuss the benefits of using cellular materials in TENGs and supercapacitors and how they can improve performance. In addition, it was reported that the challenges associated with integrating TENGs and supercapacitors and how cellular materials can address them. Overall, It is a well-written paper, and the review topic is very hot. Therefore, I recommend a minor revision of the current manuscript for publication in the Materials. The following points should be addressed: 

1.    Please provide a future direction (at least one page) section after the conclusion part. The current form is not in comprehensive form.

2.    Please add more eye-catching figures from the literature.

3.    The scheme should present the graphical abstract. It will improve the quality of the manuscript.

4.    A native speaker should check the writing style, grammar and language usage.

5.    Please ensure all Copyright permissions have been received from the original source.

6.    According to the literature, new SC materials have been reported. I kindly suggest the author add those papers and discuss for the potential TENG-SC systems. A)https://doi.org/10.1021/acs.energyfuels.2c04273 B) https://doi.org/10.1038/s41598-023-28581-5 C) https://doi.org/10.1016/j.est.2023.107392

   A native speaker should check the writing style, grammar and language usage.

Author Response

Dear reviewer:

Re: Manuscript ID: materials-2352249 and Title: The Integration of Triboelectric Nanogenerator and Supercapacitors: The Key Role of Cellular Materials

Thank you for your comments concerning our manuscript entitled “The Integration of Triboelectric Nanogenerator and Supercapacitors: The Key Role of Cellular Materials” (ID: materials-2352249). Those comments are valuable and very helpful. We have read through comments carefully and have made corrections. Based on the instructions provided in your letter, we will upload the revised manuscript.

The specific modifications are as follows:

Q1. Please provide a future direction (at least one page) section after the conclusion part.

Modification: Added section 5.4 to the text: Future direction, This provides future directions for the development of TENG-SC systems in cellular structure, including material analysis, preparation analysis, and application analysis.

Q2. Please add more eye-catching figures from the literature.

Modification: More intuitive tables and numbers have been added to the article (table 1), making it easier for readers to understand and summarize.

Q3. The scheme should present the graphical abstract. It will improve the quality of the manuscript.

Modification: The article provides a more detailed introduction to the production method and graphical abstract of the device, enabling readers to have a more intuitive understanding of the specific content of the work.

Q4. A native speaker should check the writing style, grammar and language usage.

Modification: Checked and polished the English words, grammar, and sentence structures in the article to make it more coherent and articulate.

Q5. Please ensure all Copyright permissions have been received from the original source.

Modification: We have carefully reviewed the article and ensured that all sources and references are properly cited and used in compliance with copyright laws. Additionally, we have obtained permission from the original sources where necessary.

Q6. According to the literature, new SC materials have been reported. I kindly suggest the author add those papers and discuss for the potential TENG-SC systems. A)https://doi.org/10.1021/acs.energyfuels.2c04273 B) https://doi.org/10.1038/s41598-023-28581-5 C) https://doi.org/10.1016/j.est.2023.107392

Modification: The recent work on SCs (2023) has been added to the article, which has greatly inspired TENG-SC based on its new materials and structures. The literature you mentioned is very innovative. Thank you for your valuable feedback.

We would love to thank you for allowing us to resubmit a revised copy of the manuscript and we highly appreciate your time and consideration.

Sincerely.

Reviewer 3 Report

Authors reviewed the recent advancements in the integration of TENGs and supercapacitors with cellular materials. They also explore the potential of lightweight, low-cost, and customizable cellular materials to expand the applicability of TENG-SC systems and provides insights into the development of sustainable energy harvesting and storage solutions. The overall review topic and content are good, but it needs several revisions before publication to enhance its quality and clarity.

1. The author should add a separate section discussing how cellular materials enhance the performance of TENG-SC systems. This section should provide a detailed explanation of how the porous structure and high surface area of cellular materials can improve energy generation and storage in TENG-SC systems.

2. The author should also add a section discussing recent applications of cellular materials-based TENG-SC systems, such as healthcare monitoring, self-powered sensors, powering IOT devices and catalysis.

3. The quality of the figures should be revised. For example, the arrangement of subfigures in Fig. 5 is not done properly, and there are empty spaces. Other figures such as Fig. 4a, Fig. 6b, and Fig. 7d could also be improved.

4. The introduction section requires significant revision to ensure logical clarity and to highlight the importance of the current manuscript. The author should refer to recent literature on TENGs and SCs, such as Nano Energy (2023): 108223, Chemical Engineering Journal 452 (2023): 139209, Nanomaterials 13, no. 7 (2023): 1206.

5. The author is recommended to add tables summarizing current trends, as they make it easier for readers to perceive the material and generalizations. 

Moderate editing of English language required.

Author Response

Dear reviewer:

Re: Manuscript ID: materials-2352249 and Title: The Integration of Triboelectric Nanogenerator and Supercapacitors: The Key Role of Cellular Materials

Thank you for your comments concerning our manuscript entitled “The Integration of Triboelectric Nanogenerator and Supercapacitors: The Key Role of Cellular Materials” (ID: materials-2352249). Those comments are valuable and very helpful. We have read through the comments carefully and have made corrections. Based on the instructions provided in your letter, we will upload the revised manuscript.

The specific modifications are as follows:

Q1. The author should add a separate section discussing how cellular materials enhance the performance of TENG-SC systems.

Modification: A new section titled "The Role of Cellular Materials in Enhancing the Performance of TENG-SC Systems" has been added to the manuscript in 4.3. The section discusses the advantages of using cellular structures in TENG-SC systems, including improved mechanical stability, increased surface area, and enhanced charge transfer.

Q2. The author should also add a section discussing recent applications of cellular materials-based TENG-SC systems

Modification: The latest application of TENG-SC system based on honeycomb material is added in 4.2. Including healthcare monitoring, self-powered sensors and more.

Q3. The quality of the figures should be revised.

Modification: The quality of the figures has been improved to ensure that they are clear and visually appealing for the readers. The legend numbers in the text and the numbers in the main text have been revised for accuracy and consistency.

Q4. The introduction section requires significant revision to ensure logical clarity and to highlight the importance of the current manuscript.

Modification: The introduction section has been thoroughly revised to provide a clear and logical overview of the manuscript, highlighting the importance and novelty of the research work presented in the article.

Q5. The author is recommended to add tables summarizing current trends.

Modification: Added in the text: Summary of parameters of cellular materials TENGs or SCs (Table 1).

We appreciate the opportunity to resubmit the revised manuscript and thank you for your valuable time and consideration.

Sincerely.

Round 2

Reviewer 3 Report

The manuscript has been well-revised, so I recommend it for publication.

Minor editing of English language required